# Barnyard Grass Stress Triggers Changes in Root Traits and Phytohormone Levels in Allelopathic and Non-Allelopathic Rice

**DOI:** 10.3390/biology12081074

**Published:** 2023-08-01

**Authors:** Qiling Yan, Jianhua Tong, Shuyan Li, Qiong Peng

**Affiliations:** 1Longping Branch, College of Biology, Hunan University, Changsha 410125, China; yanqiling17017@163.com (Q.Y.); lishuyan202205@163.com (S.L.); 2Hunan Provincial Key Laboratory of Phytohormones, Agricultural Biotechnology Research Institute, Hunan Academy of Agricultural Sciences, Hunan Agricultural University, Changsha 410128, China; tongjh0421@hunau.edu.cn; 3Hunan Agricultural Biotechnology Research Institute, Hunan Academy of Agricultural Sciences, Changsha 410125, China

**Keywords:** allelopathy, root trait, phytohormone, biotic stress

## Abstract

**Simple Summary:**

Barnyard grass (*Echinochloa* spp.) is one of the most dominant and noxious weed species in rice paddies. However, allelopathic rice is known to interfere with the growth of barnyard grass through allelochemical-mediated root interactions. Here, the allelopathic rice cultivar, “PI312777” (PI), and the non-allelopathic rice cultivar, “Lemont” (LE), were used as the materials for analysis and were subjected to the stresses of quinclorac-resistant (R) and -susceptible (S) barnyard grass, respectively. We first investigated changes in the morphological traits of the roots of two rice cultivars. Meanwhile, the levels of different plant hormones, including auxin, abscisic acid, jasmonic acid, and salicylic acid, known for their diverse adaptation strategies to biotic stress, were analyzed. The results indicated that PI demonstrated the greater competitive ability, compared to LE, under barnyard grass-induced stress, with respect to shoot and root biomass, the number of root tips, and root tip length in rice with root diameters of <0.5 mm. Salicylic acid (SA) and jasmonic acid (JA) were found to have the potential to suppress barnyard grass growth in allelopathic rice. The amounts of SA and JA demonstrated a significant correlation with the number of root tips and root tip length in rice with root diameters of <0.5 mm. Our research may aid in the development of strategies for reducing the environmental impact of herbicides through the prudent selection of non-chemical control tactics.

**Abstract:**

Despite the growing knowledge concerning allelopathic interference with barnyard grass, little is understood regarding the competitive physiological mechanisms of the interaction between allelopathic rice and herbicide-resistant barnyard grass. A hydroponic system was employed to investigate the root morphological traits and different phytohormonal changes in allelopathic and non-allelopathic rice cultivars when co-planted with quinclorac-resistant and -susceptible barnyard grass, respectively. The results show that shoot and root biomass were greater in PI. Barnyard grass stress induced an increase in shoot and root biomass in PI at 7 and 14 days of co-culturing rice and barnyard grass. Especially under the stress of quinclorac-resistant barnyard grass, the shoot biomass of PI increased by 23% and 68%, respectively, and the root biomass increased by 37% and 34%, respectively. In terms of root morphology, PI exhibited a significantly higher fine-root length, in root diameters of <0.5 mm, a greater number of root tips, and longer root tips compared to LE. The response to quinclorac-resistant barnyard grass stress was consistent in terms of the SA and JA content. The obvious accumulation of SA and JA was observed in two rice cultivars under quinclorac-resistant barnyard grass stress, with greater amounts of SA and JA in PI. The significant decrease in auxin (IAA) and abscisic acid (ABA) content in rice was detected from 7 to 14 days under co-culture stress. Additionally, highly significant and positive correlations were found between SA and JA content, and the number of root tips and root tip length at root diameters of 0–0.5 mm in rice.

## 1. Introduction

Barnyard grass (*Echinochloa crus-galli* L.) refers to a highly prevalent and damaging weed species that infests rice fields globally [1,2]. Barnyard grass exhibits a higher photosynthetic rate and greater utilization efficiency compared to rice [3]. The presence of barnyard grass co-culturing significantly reduces the leaf photosynthetic rate, root biomass, root oxidation activity, and dry matter accumulation of rice [4]. The worldwide rice yield losses due to barnyard grass competition were estimated to be approximately 35% [5]. As herbicides are the primary method of managing weed infestations, their prolonged and excessive use has led to the evolution of herbicide-resistant weeds [6]. The emerging and growing global incidence of herbicide-resistant weeds pose a significant threat to rice production [7,8]. Therefore, non-herbicide control tactics are crucial to reducing selection pressure for the evolution of herbicide-resistance in weeds. The importance of allelopathic interactions between rice and barnyard grass, and the mechanisms that underlie them, have been a significant area of research in non-herbicidal, alternative barnyard grass management for an extended period [9,10].

Integrated weed management research has proposed crop allelopathy as a non-herbicidal alternative [11,12]. The interference mechanism of allelopathy is characterized by the chemical suppression of neighboring plants through the release of allelochemicals from focal plants [13]. Allelopathic crop cultivars have been found to release their own ‘herbicides’ (i.e., allelochemicals) to inhibit the growth of neighboring weeds, thereby reducing the need for herbicides in paddy fields [14]. Root-secreted signaling chemicals have been found to drive plant neighbor detection and the allelochemical response [15]. The chemical relationship between allelopathic rice and weed species mainly occurs within their underground root systems. It was first discovered in 1983 that the roots of a walnut tree produce substances that inhibit weed growth [16]. In a field-based experiment in 1989, it was found that some rice varieties had an obvious inhibitive effect on weeds in neighboring paddy fields [17]. In 2013, researchers used ^13^C isotope discriminant analysis to quantitatively study the root distribution of allelopathic rice and its potential interaction with major weeds. The results showed that allelopathic rice had more and longer roots than non-allelopathic rice in the upper soil, under conditions without weeds [18]. Allelopathic rice could interfere with the growth of herbicide-resistant barnyard grass through allelochemical-mediated root interactions [19,20,21]. However, the differential responses of root traits in allelopathic rice and non-allelopathic rice to herbicide-resistant and -susceptible barnyard grass remain unclear.

The critical roles that plant hormones play in plant growth, development, and physiological responses to biotic and abiotic stresses have been extensively studied [22,23]. Biotic stress typically triggers complex changes in physiological processes in plants, such as growth changes and the synthesis of various protective compounds. In the case of systemic acquired resistance (SAR), the resulting protection is orchestrated through salicylic acid (SA) signaling, which mainly leads to protection against biotrophic pathogens [24,25]. To regulate stress-induced adaptive responses, a complex network of hormonal signals in plants interact with each other. Abscisic acid (ABA) is a major phytohormone that regulates stress responses and interacts with jasmonic acid (JA) and salicylic acid (SA) signaling pathways to allocate resources toward mitigating the effects of stressors [26]. In response to brown planthopper infection, the levels of SA and JA in rice have been found to increase simultaneously [27]. In the co-culture system of rice and barnyard grass, the SA and JA content of allelopathic rice and non-allelopathic rice have been found to be positively correlated with the allelochemicals in rice [28]. IAA controls almost all plant growth and development. IAA regulates root development and promotes disease susceptibility in plants. It inhibits SA and JA signal transduction in *Arabidopsis thaliana*, but not in rice [29,30]. Therefore, it is crucial to study changes in various plant hormones in the reaction of rice to barnyard grass stress.

In order to explore the above findings in relation to two unexplored questions, quinclorac-resistant (R) and -susceptible (S) *E. crus-galli* lines were isolated from the progeny of a single plant. These two lines were utilized to investigate the above- and belowground biomass as well as differences in root traits between allelopathic and non-allelopathic rice with barnyard grass co-culture. This aimed to further explore the dynamics of root conformation in rice in response to weed stress, which can help clarify whether the characteristics of root conformation can be exploited to enhance weed suppression. In addition, we measured the contents of four representative phytohormones in rice in response to barnyard grass stress, which provided a theoretical basis for the hormone regulation level of weed control.

## 2. Materials and Methods

### 2.1. Plant Growth and Sampling

In this study, two rice cultivars, namely ‘PI312777’ (PI) and ‘Lemont’ (LE), were utilized as experimental materials to investigate allelopathic effects. PI is a well-known allelopathic cultivar, while LE is non-allelopathic [31]. The stress treatment included quinclorac-resistant (R) and -susceptible (S) *E. crus-galli* lines, the progeny of a single plant, which had a similar genetic background. We repeated the whole experiment twice in order to ensure the authenticity of the results, and finally chose the data from one experiment to present in this manuscript. Each treatment was replicated four times in a completely randomized design. To initiate pre-germination, sterilized rice and barnyard grass seeds were separately sown onto moistened filter paper in Petri dishes (9 cm in diameter) and kept in a chamber maintained at 28 °C.

The experiment was designed to include four treatments: PI (R), LE (R), PI (S), and LE (S), where PI and LE were co-cultured with R/S barnyard grass, respectively. Additionally, PI and LE monocultures without any barnyard grass stress were considered as the controls. Seven uniform rice seedlings were transplanted into each 96-well hydroponic box (12.7 cm × 8.4 cm × 11.4 cm), which contained 1 L of Kimura B nutrient solution. The ingredients are: (NH_4_)_2_SO_4_ (365 μM), KH_2_PO_4_ (182 μM), KNO_3_ (183 μM), K_2_SO_4_ (86 μM), MgSO_4_·7H_2_O (548 μM), Ca(NO_3_)_2_·4H_2_O (366 μM), MnCl_2_·4H_2_O (9.19 μM), H_2_MoO_4_·H_2_O (0.5 μM), H_3_BO_3_ (46.2 μM), ZnSO_4_·7H_2_O (0.77 μM), CuSO_4_·5H_2_O (0.32 μM), Na_2_EDTA (20 μM), FeSO_4_·7H_2_O (20 μM), and Na_2_SiO_3_·9H_2_O (200 μM). Daily addition of distilled water was performed to maintain a consistent volume, and replacement of Kimura B nutrient solution was carried out every 7 days. At the 2-leaf stage of rice, one pre-germinated barnyard grass seedling was transplanted into the center of the hydroponic box (the co-culture ratio of rice and barnyard grass was 7:1). After 7 and 14 days of co-culturing rice and barnyard grass, the fresh weight of the shoots and roots of rice were measured after harvesting the rice seedlings. The roots were further collected to study morphological and proliferation traits.

### 2.2. Root Traits Calculation

The analysis of plant root traits was conducted according to the protocol described by Lupini et al., (2016) [32]. For root analysis, four randomly selected plants from each rice cultivar were used. The clean and undamaged roots were scanned using a Microtek ScanWizard EZ scanner (WSeen, Hangzhou, China) to produce a grayscale image. The LA-S image analysis software (Hangzhou Wanshen Detection Technology Co., Ltd., Hangzhou, China) was employed to process the image and determine the root length, root surface area, root volume in finer diameter cutoffs (0.5, 2.0, and 3.0 mm) of fine-roots, and root tip number. These morphological traits were analyzed using the LA-S software.

### 2.3. Extraction, Purification, and Determination of Plant Hormones in Rice

The analysis of phytohormones in plant tissues was carried out in accordance with a modified protocol based on Zhou et al.’s work (2017) [33]. A total of 200 mg of fresh shoot tissues were subjected to cryogenic grinding using liquid N2, and subsequently extracted with 1 mL of 80% HPLC-grade methanol (Merck, Darmstadt, Germany) at 4 °C overnight. The extracted sample was centrifuged at 15,000× *g* for 10 min, and the residue was subjected to re-extraction using 0.5 mL of 80% methanol. The supernatants obtained were vacuum freeze-dried at −60 °C and then reconstituted in 200 mL of 0.1 M sodium phosphate buffer (pH 7.8). The aqueous phase was purified using a Waters Sep-Pak C18 cartridge (Waters, MA, USA) which was washed with 800 μL of ultrapure water, and then eluted with 1.4 mL of 80% methanol. The eluate with 80% methanol was vacuum freeze-dried, and the resulting dried extract was dissolved in 40 μL of 10% methanol. This solution was then utilized for LC-MS/MS assay in a Shimadzu LC-20AD-8030 Plus MS system (Shimadzu, Kyoto, Japan).

### 2.4. Statistical Analysis

The data were presented as means ± standard error (SE) derived from four replicates for each experiment or determination. A one-way analysis of variance (ANOVA) was employed to determine any significant differences among the treatments, followed by Tukey’s honestly significant differences (HSD) tests at a significance level of *p* < 0.05. When necessary, transformations were conducted to meet the assumptions of the ANOVA. Pearson’s correlation coefficients (r) between the root morphological traits and phytohormone contents were calculated for two rice cultivars with barnyard grass at two periods. All data analyses were performed using SPSS 22.0 (SPSS Inc., Chicago, IL, USA).

## 3. Results

### 3.1. Different Reactions of Allelopathic Rice and Non-Allelopathic Rice on Shoot and Root Biomass under Barnyard Grass Stress

Due to the competition between rice and barnyard grass, differences were observed in the shoot and root biomass among allelopathic rice and non-allelopathic rice (Figure 1). Specifically, PI displayed a higher shoot and root biomass than LE at 7 and 14 days, indicating that PI showed a stronger growth advantage. Notably, barnyard grass stress resulted in a significant increase in the shoot biomass of PI in both time points. Of course, it also increased the root biomass of PI, but did not reach a significant level at 14 days. Under the stress of resistant barnyard grass, the shoot biomass of PI increased by 23% and 68%, the root biomass increased by 37% and 34%, at 7 days and 14 days, respectively. In contrast, LE exhibited an increased shoot and root biomass only under resistant barnyard grass stress, while susceptible barnyard grass stress led to a decline in the biomass. However, changes in the shoot and root biomass of LE were not statistically significant. Interestingly, PI (S) exhibited a greater shoot biomass compared to PI (R), while the opposite trend was observed for root biomass. Additionally, when exposed to susceptible barnyard grass stress, LE experienced a substantial reduction of approximately 40% in shoot biomass and 25% in root biomass after a duration of 14 days.

### 3.2. Analysis of Root Length, Root Surface Area, Root Volume, and Root Diameter of Allelopathic and Non-Allelopathic Rice under Barnyard Grass Stress

The root growth differences between two rice cultivars were illustrated through root length, root surface area, root volume, and root diameter (Figure 2). The findings indicated that PI had a notably higher total root length and slightly higher root surface area compared to LE. After 7 days of treatment, both rice cultivars showed only marginal changes in total root length with respect to their respective untreated controls. Following 14 days of co-cultivation with resistant barnyard grass, PI showed a 10% increase in total root length, while LE exhibited a 20% decrease. Conversely, after 14 days of co-cultivation with susceptible barnyard grass, total root length decreased slightly in PI, but increased in LE compared to that of single cultivation. A similar trend was observed in total root surface area for both rice cultivars at each treatment period. Resistant barnyard grass co-culture stress significantly contributed to the rise in the total root volume of PI at 14 days. Interestingly, the root mean diameter of LE was consistently higher than that of PI. However, there was no significant difference in root mean diameter between allelopathic rice and non-allelopathic rice at the seventh day of the treatment period, with only a slight change in the two rice cultivars observed after 14 days.

### 3.3. Changes in Root Tip Number and Root Tip Length of Allelopathic and Non-Allelopathic Rice under Barnyard Grass Stress

The analysis of root tip number and root tip length in three diameter ranges (from 0 to 0.5 mm; 0.5 to 2.0 mm; and 2.0 to 3.0 mm) is presented in Figure 3. The results revealed that 95–98% of the roots in both PI and LE belonged to the fine root category, with diameters < 0.5 mm. Notably, the total root tip number and root tip length of PI in the <0.5 mm diameter category were significantly higher than those of LE under the different treatments.

At 7 days, a slight decrease in root tip number was witnessed in PI, while an increase was observed in LE, compared to their respective untreated controls. However, this trend was reversed at 14 days. Under the stress of co-cultivation with resistant barnyard grass, the root tip number in PI increased and reached its peak at 14 days. Conversely, co-cultivation with both resistant and susceptible barnyard grass resulted in a 23% and 24% dip, respectively, in the root tip number of PI at 7 days. Subsequently, there was a slight increase of 2% and 0.2% in root tip number observed in PI in response to resistant and susceptible barnyard grass individually at 14 days.

Following 7 days of treatment, both PI and LE varieties showed a reduction in root tip length under barnyard grass-induced stress. The former exhibited a decrease of 9% and 14% in response to resistant and susceptible barnyard grass, respectively, compared to the controls. Additionally, the root tip length of LE also decreased under barnyard grass-induced stress. Notably, PI reached its peak root tip length at day 14 when co-cultured with resistant barnyard grass.

### 3.4. Impact of Barnyard Grass Stress on Phytohormone Levels

This experiment aimed to estimate the levels of four distinct plant hormones in allelopathic and non-allelopathic rice varieties under barnyard grass-induced stress (Figure 4). The analysis revealed a significant drop in IAA and ABA levels over time, whereas SA and JA levels showed an increase. Notably, SA levels consistently showed higher concentrations in PI (R) compared to PI and PI (S) at both time points, with the highest concentration observed in PI (R) at 14 days. Moreover, SA content was relatively higher in LE (S) than in the other treatments at 7 days. At 14 days, no significant difference was observed in SA levels among different stress conditions for LE. Similar to SA, the peak level of JA was detected in PI (R) at 14 days, exhibiting a similar pattern of changes across both rice cultivars. Compared to the control group, a reduction in IAA content was noted in PI and LE under both resistant and susceptible barnyard grass-induced stress. Meanwhile, IAA content was notably lower in PI (R) and LE (S) at 14 days. ABA content consistently remained higher in PI than LE under barnyard grass-induced stress and was reduced by barnyard grass co-culture at 7 days. However, at 14 days, ABA content was significantly higher in PI (R), and its levels were induced by resistant and susceptible stress in LE.

### 3.5. Correlation between Root Traits and Phytohormone Contents

Highly significant and positive correlations were found between SA content and root tip number (r = 0.962, *p* < 0.001) as well as root tip length (r = 0.919, *p* < 0.001) at 0–0.5 mm. However, no significant correlation was found between SA content and total root length, and among root surface area, root volume, root tip number, and root tip length at 0.5–2.0 mm (Figure 5). Similarly, JA showed a highly significant correlation with root tip number (r = 0.807, *p* < 0.01) and root tip length at 0–0.5 mm (r = 0.847, *p* < 0.001), but no significant correlation with root tip number and root tip length at 0.5–2.0 mm, as well as root surface area and root volume. In contrast, the contents of IAA and ABA had a slightly negative correlation with fine-root traits. In addition, ABA was negatively correlated with total root volume (r = −0.359, *p* < 0.05). In summary, there were significant correlations among SA and JA contents of rice with root tip number and root tip length at 0–0.5 mm.

## 4. Discussion

In the rice barnyard grass co-culture system, barnyard grass acts as a C_4_ plant, but rice is a C_3_ plant, which has a slightly competitive advantage over rice [4]. Despite the efficacy of chemical herbicides in controlling weeds, their frequent use has resulted in the selection of herbicide-resistant barnyard grass biotypes, leading to a stronger competitive edge for it. Here, in this study, we determined the above- and belowground biomass, root traits, and the four hormonal changes of allelopathic and non-allelopathic rice under the stress of barnyard grass by the co-culture of rice and barnyard grass. The selected time nodes were barnyard grass stress for 7 days and 14 days, which are the two time points with the most pronounced suppression of barnyard grass growth according to our previous experiments.

Plants with a greater phenotypic plasticity have a better chance of survival in response to environmental changes [34]. The previous studies found that the root biomass of allelopathic rice cultivar is relatively greater than that of non-allelopathic rice cultivar [18,35]. In this study, it was found that the allelopathic rice cultivar “PI312777” (PI) showed a greater competitiveness with a higher shoot and root biomass than the non-allelopathic rice cultivar “Lemont” (LE) (Figure 1). Additionally, under barnyard grass stress, PI exhibited an increased shoot and root biomass, LE showed a similar trend in response to resistant barnyard grass stress, whereas the figures were decreased under susceptible barnyard grass stress. These findings suggested that the allelopathic rice cultivar might pose a stronger neighbor-identity recognition ability in response to barnyard grass stress, compared to the non-allelopathic rice cultivar. Notably, we observed more pronounced changes in the shoot and root biomass of PI when co-cultured with barnyard grass, indicating the role of chemically mediated signaling interactions between allelopathic crop and competing weeds in influencing crop growth. Plant interactions within mixed-species systems were shaped by the competition for growth resources both above and below ground, with belowground interactions having a more significant impact on the performance of coexisting plants than aboveground interactions [36]. This study found that the shoot biomass of PI (S) was greater than that of PI (R), while the root biomass showed an entirely opposite trend. Thus, we hypothesize that allelopathic rice cultivars may confer a competitive advantage for underground growth under resistant barnyard grass stress.

Belowground ecological interactions greatly affected crop–weed allelopathic action [37]. The development and interaction of roots played crucial roles in the interference of allelopathic rice with barnyard grass [14]. Root traits were significantly positively correlated to allelopathic inhibition in rice root exudates [38]. Our study found that PI exhibited a significantly higher fine-root length, more root tip number, and a greater root tip length compared to LE (Figure 2 and Figure 3), in agreement with previously reported results by Li et al., (2019) [36]. After 14 days of co-culture (Figure 2), the total root surface and root volume of PI were significantly induced by resistant barnyard grass, coupled with the greatest root tip length, at this time point. Consequently, we hypothesize that the reason for this is that PI accumulates and releases more allelochemicals into the culture solution, resulting in a high inhibition of the target plant. Of course, further experimental investigation of the substances in PI is still needed. Smaller diameter cutoffs of root (e.g., 1.0 or 0.5 mm) were regarded as more functional fine roots [39]. The study revealed that 95–98% of the roots of both PI and LE had fine roots with a diameter of less than 0.5 mm. In addition, the total number of root tips and root tip length with a diameter of less than 0.5 mm in PI was higher than that in LE across different treatments (Figure 3). There could be significant correlations between the allelopathic potential of rice and the length of fine-root tips with a diameter of 0–0.5 mm.

The ubiquitous signaling chemicals, SA and JA, have been shown to elicit the production of defensive plant metabolites against microbes, herbivores, or competitors [40,41]. In our study, the contents of both SA and JA were significantly increased between 7 and 14 days (Figure 4A,B). And the levels of SA and JA were significantly higher in PI co-cultivated with resistant barnyard grass compared to PI alone or PI co-cultivated with susceptible barnyard grass. The reason might be that resistant barnyard grass has a stronger resistance and growth potential than susceptible barnyard grass and caused more stressful effects on rice, so both of them in PI (R) were higher. These results indicated that the SA- and JA-dependent defense mechanisms were in response to barnyard grass stress. The most potent allelopathic ability of PI was observed at 14 days, as evidenced by the highest SA and JA contents in PI (R). This finding was consistent with the results of root traits in allelopathic rice in the presence of resistant barnyard grass. The data generated in this study indicated that the levels of IAA and ABA tended to drop from 7 to 14 days (Figure 4C,D). The authors hypothesized that the allelopathic effects on barnyard grass were inversely correlated with the contents of IAA and ABA. Auxin not only played a crucial role in plant growth and development but was also closely associated with plant biotic stress [22,42]. After SAR induction, a lot of auxin-responsive genes were also repressed, clearly indicating that auxin promotes disease susceptibility, and that an enhanced resistance to disease requires the repression of auxin signaling [30]. The barnyard grass stress resulted in a decrease in IAA content in both rice materials, except in LE (R) at 14 days. We speculated that auxin signaling was inhibited when rice actively responds to barnyard grass stress, and then the auxin content was decreased. Moreover, the experiment revealed the response variation rules of rice ABA content under barnyard grass stress, likely due to the antagonistic role of ABA and SA signaling pathways in rice [43]. Through correlation analysis, we found that SA and JA in the shoots of rice seedlings were significantly correlated with the fine-root traits of the root tip numbers and root tip lengths of rice at 0–0.5 mm (Figure 5), indicating that SA and JA had a potential positive regulation role to suppress barnyard grass growth in allelopathic rice.

## 5. Conclusions

This study revealed that the allelopathic rice cultivar “PI” demonstrated a greater competitive ability with respect to shoot and root biomass as compared to the non-allelopathic rice cultivar “LE” under conditions of barnyard grass stress. Meanwhile, allelopathic rice cultivar exhibited a significantly greater length of fine roots with a diameter of less than 0.5 mm, an increased number of root tips, and longer root tips, compared to the non-allelopathic rice cultivar co-cultivated with barnyard grass in a hydroponic system. The results also demonstrated significant correlations between rice allelopathic potential and rice fine-root tip length at 0–0.5 mm in diameter. Additionally, barnyard grass stress triggered SA- and JA-dependent defense responses. The contents of SA and JA in the shoots of rice seedlings were significantly positively correlated with root tip number and root tip length at 0–0.5 mm. To illustrate the regulating effects of phytohormone-based interactions in improving the ability of rice to resistant barnyard grass stress, further study is essential to analyze the SA- and JA-related gene expression and enzyme activity.

## Figures and Tables

**Figure 1 biology-12-01074-f001:**
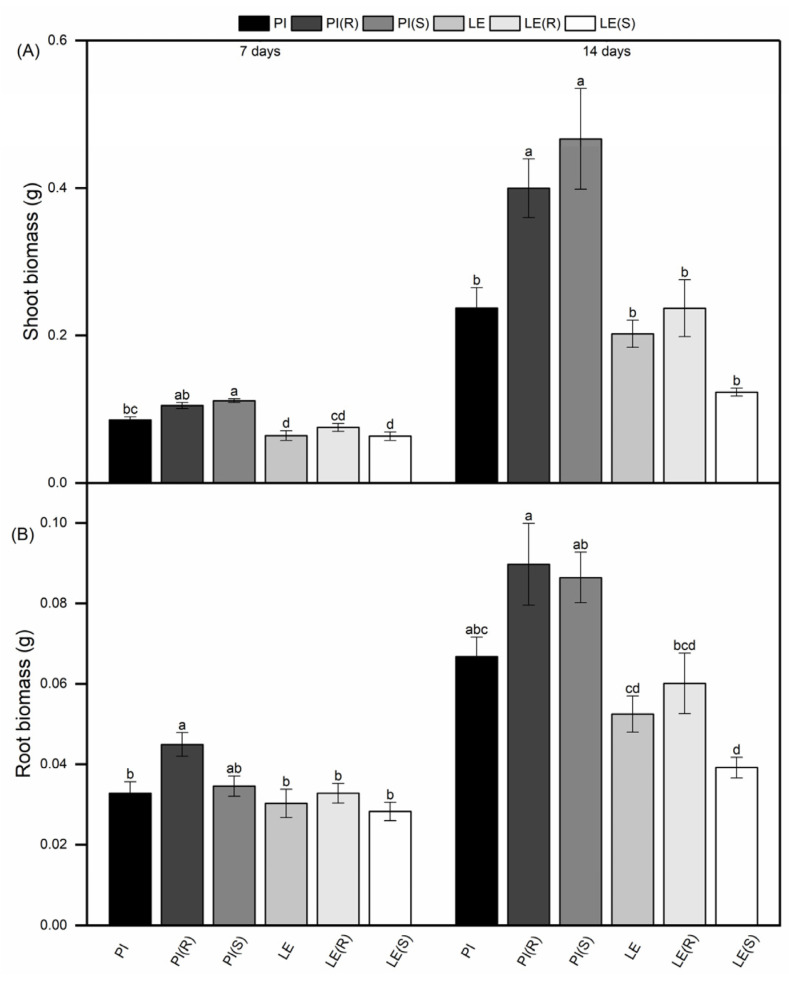
Shoot biomass (**A**) and root biomass (**B**) of two rice cultivars in different planting patterns (PI monoculture; LE monoculture; (R): co-cultured with resistant barnyard grass; (S): co-cultured with susceptible barnyard grass). Each value was the mean ± SE. Bars with different letters were significantly different at the *p* < 0.05 level according to Tukey’s test. The results obtained were in line with those shown below.

**Figure 2 biology-12-01074-f002:**
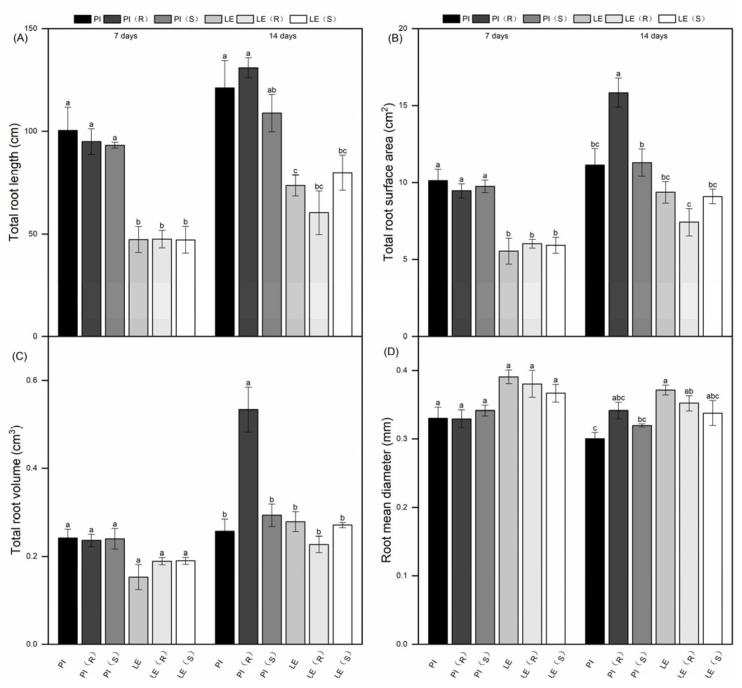
Effect of quinclorac-resistant and -susceptible barnyard grass on total root length (**A**), total root surface area (**B**), total root volume (**C**), and total root mean diameter (**D**) of allelopathic and non-allelopathic rice cultivars. Bars with different letters were significantly different at the *p* < 0.05 level according to Tukey’s test. The results obtained were in line with those shown below.

**Figure 3 biology-12-01074-f003:**
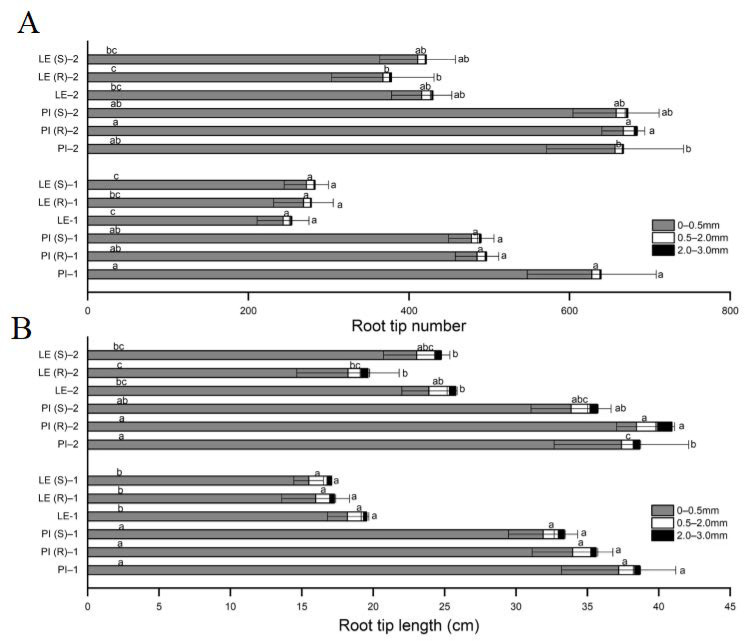
Total root tip number (**A**) and root tip length (**B**) in the fine-root diameter range of 0–0.5, 0.5–2.0, and 2.0–3.0 mm of allelopathic and non-allelopathic rice cultivars. ‘−1’ and ‘−2’ represent the 7 and 14 days after the different treatments, respectively. Bars with different letters were significantly different at the *p* < 0.05 level according to Tukey’s test. The results obtained were in line with those shown below.

**Figure 4 biology-12-01074-f004:**
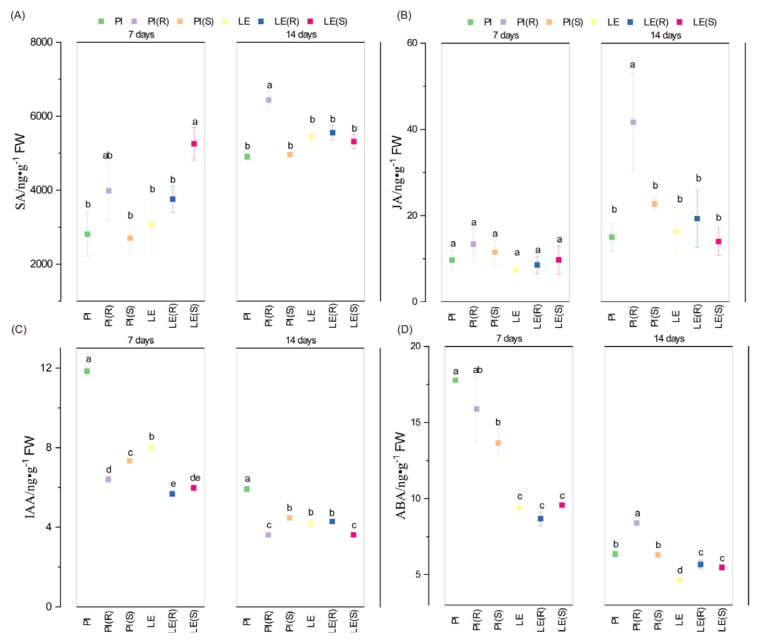
Differential levels of representative phytohormones in allelopathic rice and non-allelopathic rice cultivars, including salicylic acid (**A**), jasmonic acid (**B**), auxin (**C**), and abscisic acid (**D**). Bars with different letters were significantly different at the *p* < 0.05 level according to Tukey’s test. The results obtained were in line with those shown below.

**Figure 5 biology-12-01074-f005:**
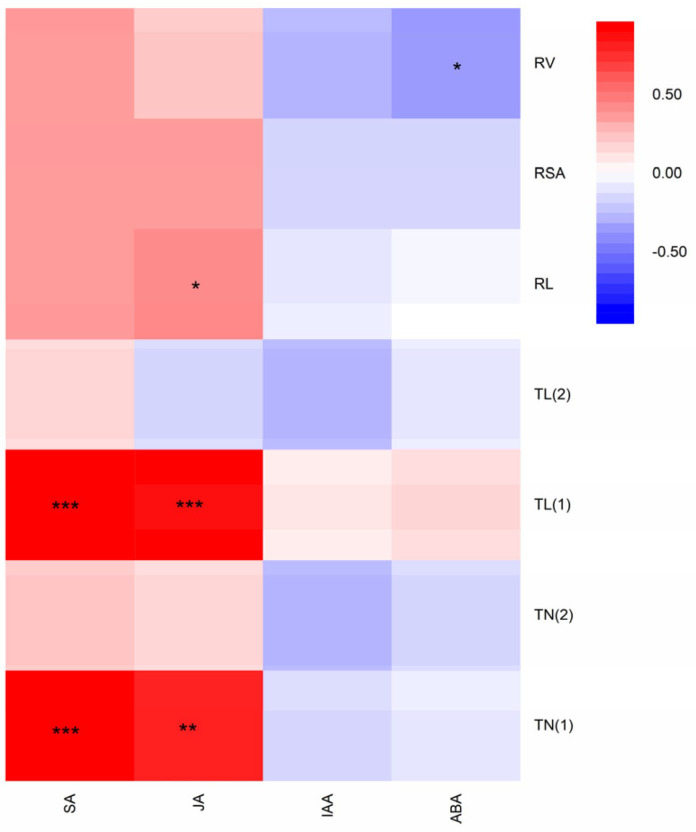
Correlation coefficients (r) among fine-root traits and phytohormone contents of allelopathic and non-allelopathic rice. TN (1)—root tip number at 0–0.5 mm; TN (2)—root tip number at 0.5–2.0 mm; TL (1)—root tip length at 0–0.5 mm; TL (2)—root tip length at 0.5–2.0 mm; RL—root length, RSA—root surface area, RV—root volume, * 0.01 < *p* < 0.05; ** 0.001 < *p* < 0.01; *** *p* < 0.001.

## Data Availability

Data supporting the results are available by contacting the corresponding author.

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
