# Peer review of "Barnyard Grass Stress Triggers Changes in Root Traits and Phytohormone Levels in Allelopathic and Non-Allelopathic Rice"

_biology, 2023, doi:10.3390/biology12081074_

Round 1

Reviewer 1 Report

Overall, this is a good manuscript that needs improvements. The introduction is relevant, but as a suggestion, it should contain some information about Indole acetic acid (IAA) and the objective(s) of the manuscript. Sufficient information about Material and Methods is presented for readers to follow the present study. The Results are generally appropriate, although clarification of a few details is needed. The Discussion needs improvement to clarify why quinclorac-resistant barnyardgrass induces up-regulation of SA and JA. I have listed some suggestions for manuscript improvement (document attached).

Minor edits in English are needed.

Reviewer 2 Report

Comments to the Author
The topic of the paper is interesting and covers the goals of the journal. One of the main concerns I have is the lack of a second run (repetition) of the experiment.

Taking into account the fact that the time of weed seed germination and emergence changes the ability of weeds to affect crops and that weeds usually grow faster than crops, so why pre-germinated barnyardgrass seedling was transplanted at the 2-leaf stage of rice? If the simulation of transplanted rice farming is intended (not direct seeded), the 2-leaf stage is also not the stage that farmers do transplanting.

Minor comments

Line 26: the ‘ herbicide in ” non-chemical herbicide”, is redundant

Line 47: change “rice cultivation” to “rice fields”

In Fig.1 please recheck the "letter of mean comparisons" for LE(S), (14 day for shoot).

Line 227: Since the authors evaluated only two rice cultivars, I think there is no need to analyze the correlation between root traits and plant hormone content. It is usually reasonable to perform this type of analysis if there are a large number of cultivars.

 Line 245-254: I think this paragraph is suitable for introduction section. In the “Discussion’ author should discuss their results.

Reviewer 3 Report

The manuscript entitles 'Barnyardgrass stress triggers root traits and phytohormones levels changes in allelopathic and non-allelopathic rice. The manuscript is well-written and the study is very interesting. The manuscript needs the following revisions:

Line 22 - Please define SA and JA before using the abbreviations. 

Line 39 - Please define IAA and ABA 

Abstract- If possible, please add some comparison numbers or percentages to show how much increase in shoot and root biomass was observed between PI and LE cultivars. Some numbers of comparison will add more to the abstract. 

Line 65-77 - Please check the grammar of the sentence or rewrite it. 

Introduction - I recommend adding further information on the PI and LE rice cultivars and if possible citing other studies that have used these cultivars or different allelopathic rice cultivars. Mention studies that show the PI cultivar as a well-known allelopathic cultivar. Further information on quinclorac-resistant barnyardgrass will be helpful. 

Line 88 - Please cite this. 

Line 89-90 - Needs more explanation on how stress treatment was derived from a single plant. Please add why a hydroponic system was selected instead of soil or other substrate. 

Line 99 - Please add the nutrients present in the nutrient solution. 

Discussion - I recommend explaining why this study was only done for a duration of 7 and 14 days. And whether or not results would vary with added factors of soil, microbes, and other soil/substrate chemistry. 

Needs some revision in the introduction. Minor grammatical errors. 
